# A Retrieve-and-Edit Framework for Predicting Structured Outputs

**Tatsunori B. Hashimoto**
Department of Computer Science
Stanford University
thashim@stanford.edu

**Kelvin Guu**
Department of Statistics
Stanford University
kguu@stanford.edu

**Yonatan Oren**
Department of Computer Science
Stanford University
yonatano@stanford.edu

**Percy Liang**
Department of Computer Science
Stanford University
pliang@cs.stanford.edu

## Abstract

For the task of generating complex outputs such as source code, editing existing outputs can be easier than generating complex outputs from scratch. With this motivation, we propose an approach that first retrieves a training example based on the input (e.g., natural language description) and then edits it to the desired output (e.g., code). Our contribution is a computationally efficient method for learning a retrieval model that embeds the input in a task-dependent way without relying on a hand-crafted metric or incurring the expense of jointly training the retriever with the editor. Our retrieve-and-edit framework can be applied on top of any base model. We show that on a new autocomplete task for GitHub Python code and the Hearthstone cards benchmark, retrieve-and-edit significantly boosts the performance of a vanilla sequence-to-sequence model on both tasks.

## 1 Introduction

In prediction tasks with complex outputs, generating well-formed outputs is challenging, as is well-known in natural language generation [20, 28]. However, the desired output might be a variation of another, previously-observed example [14, 13, 30, 18, 24]. Other tasks ranging from music generation to program synthesis exhibit the same phenomenon: many songs borrow chord structure from other songs, and software engineers routinely adapt code from Stack Overflow.

Motivated by these observations, we adopt the following *retrieve-and-edit* framework (Figure 1):

1. **Retrieve:** Given an input $x$, e.g., a natural language description 'Sum the first two elements in `tmp`', we use a *retriever* to choose a similar training example $(x', y')$, such as 'Sum the first 5 items in `Customers`'.

2. **Edit:** We then treat $y'$ from the retrieved example as a "prototype" and use an *editor* to edit it into the desired output $y$ appropriate for the input $x$.

While many existing methods combine retrieval and editing [13, 30, 18, 24], these approaches rely on a fixed hand-crafted or generic retrieval mechanism. One drawback to this approach is that designing a task-specific retriever is time-consuming, and a generic retriever may not perform well on tasks where $x$ is structured or complex [40]. Ideally, the retrieval metric would be learned from the data in a task-dependent way: we wish to consider $x$ and $x'$ similar only if their corresponding *outputs* $y$ and $y'$ differ by a small, easy-to-perform edit. However, the straightforward way of training a retriever

jointly with the editor would require summing over all possible $x'$ for each example, which would be prohibitively slow.

In this paper, we propose a way to train a retrieval model that is optimized for the downstream edit task. We first train a noisy encoder-decoder model, carefully selecting the noise and embedding space to ensure that inputs that receive similar embeddings can be easily edited by an oracle editor. We then train the editor by retrieving according to this learned metric. The main advantage of this approach is that it is computationally efficient and requires no domain knowledge other than an encoder-decoder model with low reconstruction error.

We evaluate our retrieve-and-edit approach on a new Python code autocomplete dataset of 76k functions, where the task is to predict the next token given partially written code and natural language description. We show that applying the retrieve-and-edit framework to a standard sequence-to-sequence model boosts its performance by 14 points in BLEU score [25]. Comparing retrieval methods, learned retrieval improves over a fixed, bag-of-words baseline by 6 BLEU. We also evaluate on the Hearthstone cards benchmark [22], where systems must predict a code snippet based on card properties and a natural language description. We show that augmenting a standard sequence-to-sequence model with the retrieve-and-edit approach improves the model by 7 BLEU and outperforms the best non-abstract syntax tree (AST) based model by 4 points.

## 2 Problem statement

**Task.** Our goal is to learn a model $p_{\mathrm{model}}(y \mid x)$ that predicts an output $y$ (e.g., a 5–15 line code snippet) given an input $x$ (e.g., a natural language description) drawn from a distribution $p_{\mathrm{data}}$. See Figure 1 for an illustrative example.

**Retrieve-and-edit.** The retrieve-and-edit framework corresponds to the following generative process: given an input $x$, we first retrieve an example $(x', y')$ from the training set $\mathcal{D}$ by sampling using a *retriever* of the form $p_{\mathrm{ret}}((x', y') \mid x)$. We then generate an output $y$ using an *editor* of the form $p_{\mathrm{edit}}(y \mid x, (x', y'))$. The overall likelihood of generating $y$ given $x$ is

$$p_{\mathrm{model}}(y \mid x) = \sum_{(x', y') \in \mathcal{D}} p_{\mathrm{edit}}(y \mid x, (x', y')) p_{\mathrm{ret}}((x', y') \mid x), \tag{1}$$

and the objective that we seek to maximize is

$$\mathcal{L}(p_{\mathrm{edit}}, p_{\mathrm{ret}}) := \mathbb{E}_{(x,y) \sim p_{\mathrm{data}}} \left[\log p_{\mathrm{model}}(y \mid x)\right]. \tag{2}$$

For simplicity, we focus on deterministic retrievers, where $p_{\mathrm{ret}}((x', y') \mid x)$ is a point mass on a particular example $(x', y')$. This matches the typical approach for retrieve-and-edit methods, and we leave extensions to stochastic retrieval [14] and multiple retrievals [13] to future work.

**Learning task-dependent similarity.** As mentioned earlier, we would like the retriever to incorporate *task-dependent similarity*: two inputs $x$ and $x'$ should be considered similar only if the editor has a high likelihood of editing $y'$ into $y$. The optimal retriever for a fixed editor would be one that maximizes the standard maximum marginal likelihood objective in equation (1).

An initial idea to learn the retriever might be to optimize for maximum marginal likelihood using standard approaches such as gradient descent or expectation maximization (EM). However, both of

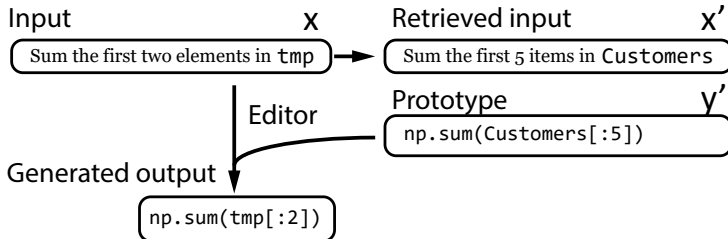

**Figure 1.** The retrieve-and-edit approach consists of the retriever, which identifies a relevant example from the training set, and the editor, which predicts the output conditioned on the retrieved example.

these approaches involve summing over all training examples $\mathcal{D}$ on each training iteration, which is computationally intractable.

Instead, we break up the optimization problem into two parts. We first train the retriever in isolation, replacing the edit model $p_{\text{edit}}$ with an *oracle* editor $p_{\text{edit}}^*$ and optimizing a lower bound for the marginal likelihood under this editor. Then, given this retriever, we train the editor using the standard maximum likelihood objective. This decomposition makes it possible to avoid the computational difficulties of learning a task-dependent retrieval metric, but importantly, we will still be able to learn a retriever that is task-dependent.

# 3 Learning to retrieve and edit

We first describe the procedure for training our retriever (Section 3.1), which consists of embedding the inputs $x$ into a vector space (Section 3.1.1) and retrieving according to this embedding. We then describe the editor and its training procedure, which follows immediately from maximizing the marginal likelihood (Section 3.2).

## 3.1 Retriever

Sections 3.1.1–3.1.3 will justify our training procedure as maximization of a lower bound on the likelihood; one can skip to Section 3.1.4 for the actual training procedure if desired.

We would like to train the retriever based on $\mathcal{L}$ (Equation 2), but we do not yet know the behavior of the editor. We can avoid this problem by optimizing the retriever $p_{\text{ret}}$, assuming the editor is the true conditional distribution over the targets $y$ given the retrieved example $(x', y')$ under the joint distribution $p_{\text{ret}}((x', y') \mid x) p_{\text{data}}(x, y)$. We call this the *oracle editor* for $p_{\text{ret}}$,

$$p_{\text{edit}}^*(y \mid (x', y')) = \frac{\sum_x p_{\text{ret}}((x', y') \mid x) p_{\text{data}}(x, y)}{\sum_{x,y} p_{\text{ret}}((x', y') \mid x) p_{\text{data}}(x, y)}.$$

The oracle editor gives rise to the following lower bound on $\sup_{p_{\text{edit}}} \mathcal{L}(p_{\text{ret}}, p_{\text{edit}})$

$$\mathcal{L}^*(p_{\text{ret}}) := \mathbb{E}_{(x,y) \sim p_{\text{data}}}[\mathbb{E}_{(x',y') \mid x \sim p_{\text{ret}}}[\log p_{\text{edit}}^*(y \mid (x', y'))]], \tag{3}$$

which follows from Jensen's inequality and using a particular editor $p_{\text{edit}}^*$ rather than the best possible $p_{\text{edit}}$.[1] Unlike the real editor $p_{\text{edit}}$, $p_{\text{edit}}^*$ does not condition on the input $x$ to ensure that the bound represents the quality of the retrieved example alone.

Next, we wish to find a further lower bound that takes the form of a distance minimization problem:

$$\mathcal{L}^*(p_{\text{ret}}) \geq C - \mathbb{E}_{x \sim p_{\text{data}}}[\mathbb{E}_{x' \mid x \sim p_{\text{ret}}}[d(x', x)^2]], \tag{4}$$

where $C$ is a constant independent of $p_{\text{ret}}$. The $p_{\text{ret}}$ that maximizes this lower bound is the deterministic retriever which finds the nearest neighbor to $x$ under the metric $d$.

In order to obtain such a lower bound, we will learn an encoder $p_\theta(v \mid x)$ and decoder $p_\phi(y \mid v)$ and use the distance metric in the latent space of $v$ as our distance $d$. When $p_\theta(v \mid x)$ takes a particular form, we can show that this results in the desired lower bound (4).

### 3.1.1 The latent space as a task-dependent metric

Consider any encoder-decoder model with a probabilistic encoder $p_\theta(v \mid x)$ and decoder $p_\phi(y \mid v)$. We can show that there is a variational lower bound that takes a form similar to (4) and decouples $p_{\text{ret}}$ from the rest of the objective.

**Proposition 1.** *For any densities $p_\theta(v \mid x)$ and $p_\phi(y \mid v)$ and random variables $(x, y, x', y') \sim p_{ret}((x', y') \mid x) p_{data}(x, y)$,*

$$\mathcal{L}^*(p_{\text{ret}}) \geq \underbrace{\mathbb{E}_{(x,y) \sim p_{data}}[\mathbb{E}_{v \sim p_\theta(v|x)}[\log p_\phi(y \mid v)]]}_{:=\mathcal{L}_{reconstruct}(\theta,\phi)} - \underbrace{\mathbb{E}_x[\mathbb{E}_{x' \mid x \sim p_{ret}}[KL(p_\theta(v \mid x) \| p_\theta(v \mid x'))]]}_{:=\mathcal{L}_{discrepancy}(\theta,p_{ret})}. \tag{5}$$

**Proof** The inequality follows from standard arguments on variational approximations. Since $p^*_{\text{edit}}(y \mid (x', y'))$ is the conditional distribution implied by the joint distribution $(x', y', x, y)$, we have:

$$\mathbb{E}_{y|x',y'\sim p^*_{\text{edit}}}[\log p^*_{\text{edit}}(y \mid (x', y'))] \geq \mathbb{E}_{y|x',y'\sim p^*_{\text{edit}}}\left[\log \int p_\phi(y \mid v)p_\theta(v \mid x')dv\right],$$

where $\int p_\phi(y \mid v)p_\theta(v \mid x')dv$ is just another distribution. Taking the expectation of both sides with respect to $(x, x', y')$ and applying law of total expectation yields:

$$\mathcal{L}^*(p_{\text{ret}}) \geq \mathbb{E}_{(x,y)\sim p_{\text{data}}}\left[\mathbb{E}_{(x',y')|x\sim p_{\text{ret}}}\left[\log \int p_\phi(y \mid v)p_\theta(v \mid x')dv\right]\right]. \tag{6}$$

Next, we apply the standard evidence lower bound (ELBO) on the latent variable $v$ with variational distribution $p_\theta(v \mid x)$. This continues the lower bounds

$$\geq \mathbb{E}_{(x,y)\sim p_{\text{data}}}\left[\mathbb{E}_{(x',y')|x\sim p_{\text{ret}}}\left[\mathbb{E}_{v|x\sim p_\theta}\left[\log p_\phi(y \mid v)\right] - \text{KL}(p_\theta(v \mid x)\|p_\theta(v \mid x'))\right]\right]$$

$$\geq \mathbb{E}_{(x,y)\sim p_{\text{data}}}[\mathbb{E}_{v|x\sim p_\theta}[\log p_\phi(y \mid v)]] - \mathbb{E}_{x\sim p_{\text{data}}}[\mathbb{E}_{x'\sim p_{\text{ret}}}[\text{KL}(p_\theta(v \mid x)\|p_\theta(v \mid x'))]],$$

where the last inequality is just collapsing expectations. □

Proposition 1 takes the form of the desired lower bound (4), since it decouples the reconstruction term $\mathbb{E}_{(x,y)\sim p_{\text{data}}}\left[\mathbb{E}_{v|x\sim p_\theta}\left[\log p_\phi(y \mid v)\right]\right]$ from a discrepancy term $\text{KL}(p_\theta(v \mid x)\|p_\theta(v \mid x'))$. However, there are two differences between the earlier lower bound (4) and our derived result. The KL-divergence may not represent a distance metric, and there is dependence on unknown parameters $(\theta, \phi)$. We will resolve these problems next.

### 3.1.2 The KL-divergence as a distance metric

We will now show that for a particular choice of $p_\theta$, the KL divergence $\text{KL}(p_\theta(v \mid x)\|p_\theta(v \mid x'))$ takes the form of a squared distance metric. In particular, choose $p_\theta(v \mid x)$ to be a von Mises-Fisher distribution over unit vectors centered on the output of an encoder $\mu_\theta(x)$:

$$p_\theta(v \mid x) = \text{vMF}(v; \mu_\theta(x), \kappa) = C_\kappa \exp\left(\kappa\mu_\theta(x)^\top v\right), \tag{7}$$

where both $v$ and $\mu_\theta(x)$ are unit vectors, and $C_\kappa$ is a normalization constant depending only on $d$ and $\kappa$. The von Mises-Fisher distribution $p_\theta$ turns the KL divergence term into a squared Euclidean distance on the unit sphere (see the Appendix A). This further simplifies the discrepancy term (5) to

$$\mathcal{L}_{\text{discrepancy}}(\theta, p_{\text{ret}}) = C_\kappa \mathbb{E}_{x\sim p_{\text{data}}}[\mathbb{E}_{x'\sim p_{\text{ret}}}[\|\mu_\theta(x) - \mu_\theta(x')\|_2^2]], \tag{8}$$

The KL divergence on other distributions such as the Gaussian can also be expressed as a distance metric, but we choose the von-Mises Fisher since the KL divergence is upper bounded by a constant, a property that we will use next.

The retriever $p_{\text{ret}}$ that minimizes (8) deterministically retrieves the $x'$ that is closest to $x$ according to the embedding $\mu_\theta$. For efficiency, we implement this retriever using a cosine-LSH hash via the `annoy` Python library, which we found to be both accurate and scalable.

### 3.1.3 Setting the encoder-decoder parameters $(\theta, \phi)$

Any choice of $(\theta, \phi)$ turns Proposition 1 into a lower bound of the form (4), but the bound can potentially be very loose if these parameters are chosen poorly. Joint optimization over $(\theta, \phi, p_{\text{ret}})$ is computationally expensive, as it requires a sum over the potential retrieved examples. Instead, we will optimize $\theta, \phi$ with respect to a conservative lower bound that is independent of $p_{\text{ret}}$. For the von-Mises Fisher distribution, $\text{KL}(p_\theta(v \mid x)\|p_\theta(v \mid x')) \leq 2C_\kappa$, and thus

$$\mathbb{E}_{(x,y)\sim p_{\text{data}}}[\mathbb{E}_{v|x\sim p_\theta}[\log p_\phi(y \mid v)]] - \mathbb{E}_{x\sim p_{\text{data}}}[\mathbb{E}_{x'\sim p_{\text{ret}}}[\text{KL}(p_\theta(v \mid x)\|p_\theta(v \mid x'))]]$$

$$\geq \mathbb{E}_{(x,y)\sim p_{\text{data}}}[\mathbb{E}_{v|x\sim p_\theta}[\log p_\phi(y \mid v)]] - 2C_\kappa.$$

Therefore, we can optimize $\theta, \phi$ with respect to this worst-case bound. This lower bound objective is analogous to the recently proposed hyperspherical variational autoencoder and is straightforward to train using reparametrization gradients [9, 14, 38]. Our training procedure consists of applying minibatch stochastic gradient descent on $(\theta, \phi)$ where gradients involving $v$ are computed with the reparametrization trick.

### 3.1.4 Overall procedure

The overall retrieval training procedure consists of two steps:

1. Train an encoder-decoder to map each input $x$ into an embedding $v$ that can reconstruct the output $y$:

$$(\hat{\theta}, \hat{\phi}) := \arg\max_{\theta, \phi} \mathbb{E}_{(x,y) \sim p_{\text{data}}}[\mathbb{E}_{v|x \sim p_\theta}[\log p_\phi(y \mid v)]]. \tag{9}$$

2. Set the retriever to be the deterministic nearest neighbor input in the training set under the encoder:

$$\hat{p}_{\text{ret}}(x', y' \mid x) := \mathbf{1}[(x', y') = \arg\min_{(x',y') \in \mathcal{D}} \|\mu_{\hat{\theta}}(x) - \mu_{\hat{\theta}}(x')\|_2^2]. \tag{10}$$

### 3.2 Editor

The procedure in Section 3.1.4 returns a retriever $\hat{p}_{\text{ret}}$ that maximizes a lower bound on $\mathcal{L}^*$, which is defined in terms of the oracle editor $p_{\text{edit}}^*$. Since we do not have access to the oracle editor $p_{\text{edit}}^*$, we train the editor $p_{\text{edit}}$ to directly maximize $\mathcal{L}(p_{\text{edit}}, \hat{p}_{\text{ret}})$.

Specifically, we solve the optimization problem:

$$\arg\max_{p_{\text{edit}}} \mathbb{E}_{(x,y) \sim p_{\text{data}}}[\mathbb{E}_{(x',y') \sim \hat{p}_{\text{ret}}}[\log p_{\text{edit}}(y \mid x, (x', y'))]]. \tag{11}$$

In our experiments, we let $p_{\text{edit}}$ be a standard sequence-to-sequence model with attention and copying [12, 36] (see Appendix B for details), but any model architecture can be used for the editor.

## 4 Experiments

We evaluate our retrieve-and-edit framework on two tasks. First, we consider a code autocomplete task over Python functions taken from GitHub and show that retrieve-and-edit substantially outperforms approaches based only on sequence-to-sequence models or retrieval. Then, we consider the Hearthstone cards benchmark and show that retrieve-and-edit can boost the accuracy of existing sequence-to-sequence models.

For both experiments, the dataset is processed by standard space-and-punctuation tokenization, and we run the retrieve and edit model with randomly initialized word vectors and $\kappa = 500$, which we obtained by evaluating BLEU scores on the development set of both datasets. Both the retriever and editor were trained for 1000 iterations on Hearthstone and 3000 on GitHub via ADAM minibatch gradient descent, with batch size 16 and a learning rate of 0.001.

### 4.1 Autocomplete on Python GitHub code

Given a natural language description of a Python function and a partially written code fragment, the task is to return a candidate list of $k = 1, 5, 10$ next tokens (Figure 2). A model predicts correctly if the ground truth token is in the candidate list. The performance of a model is defined in terms of the average or maximum number of successive tokens correctly predicted.

**Dataset.** Our Python autocomplete dataset is a representative sample of Python code from GitHub, obtained from Google Bigquery by retrieving Python code containing at least one block comment with restructured text (reST) formatting (See Appendix C for details). We use this data to form a code prediction task where each example consists of four inputs: the block comment, function name, arguments, and a partially written function body. The output is the next token in the function body.

To avoid the possibility that repository forks and duplicated library files result in a large number of duplicate functions, we explicitly deduplicated all files based on both the file contents and repository path name. We also removed any duplicate function/docstring pairs and split the train and test set at the repository level. We tokenized using space and punctuation and kept only functions with at most 150 tokens, as the longer functions are nearly impossible to predict from the docstring. This resulted in a training set of 76k Python functions.

| | Longest completed length | | | Avg completion length | | | BLEU |
|---|---|---|---|---|---|---|---|
| | k=1 | k=5 | k=10 | k=1 | k=5 | k=10 | |
| Retrieve-and-edit (Retrieve+Edit) | **17.6** | **20.9** | **21.9** | **5.8** | **7.5** | **8.1** | **34.7** |
| Seq2Seq | 10.6 | 12.5 | 13.2 | 2.5 | 3.4 | 3.8 | 19.2 |
| Retriever only (TaskRetriever) | 13.5 | | | 4.7 | | | 29.9 |

**Table 1.** Retrieve-and-edit substantially improves the performance over baseline sequence-to-sequence models (Seq2Seq) and trained retrieval without editing (TaskRetriever) on the Python autocomplete dataset. $k$ indicates the number of candidates over beam-search considered for predicting a token, and completion length is the number of successive tokens that are correctly predicted.

| | Longest completed length | Avg completion length | BLEU |
|---|---|---|---|
| TaskRetriever | 13.5 | 4.7 | 29.9 |
| InputRetriever | 12.3 | 4.1 | 29.8 |
| LexicalRetriever | 9.8 | 3.4 | 23.1 |

**Table 2.** Retrievers based on the noisy encoder-decoder (TaskRetriever) outperform a retriever based on bag-of-word vectors (LexicalRetriever). Learning an encoder-decoder on the inputs alone (InputRetriever) results in a slight loss in accuracy.

**Results.** Comparing the retrieve-and-edit model (Retrieve+Edit) to a sequence-to-sequence baseline (Seq2Seq) whose architecture and training procedure matches that of the editor, we find that retrieval adds substantial performance gains on all metrics with no domain knowledge or hand-crafted features (Table 1).

We also evaluate various retrievers: TaskRetriever, which is our task-dependent retriever presented in Section 3.1; LexicalRetriever, which embeds the input tokens using a bag-of-word vectors and retrieves based on cosine similarity; and InputRetriever, which uses the same encoder-decoder architecture as TaskRetriever but modifies the decoder to predict $x$ rather than $y$. Table 2 shows that TaskRetriever significantly outperforms LexicalRetriever on all metrics, but is comparable to InputRetriever on BLEU and slightly better on the autocomplete metrics. We did not directly compare to abstract syntax tree (AST) based methods here since they do not have a direct way to condition on partially-generated code, which is needed for autocomplete.

Examples of predicted outputs in Figure 2 demonstrate that the docstring does not fully specify the structure of the output code. Despite this, the retrieval-based methods are sometimes able to retrieve relevant functions. In the example, the retriever learns to return a function that has a similar conditional check. Retrieve+Edit does not have enough information to predict the true function and therefore predicts a generic conditional (if not b_data). In contrast, the seq2seq defaults to predicting a generic getter function rather than a conditional.

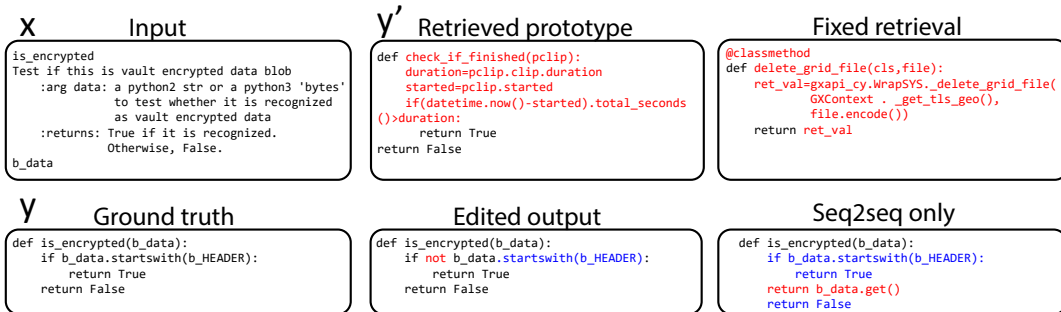

**Figure 2.** Example from the Python autocomplete dataset along with the retrieved example used during prediction (top center) and baselines (right panels). The edited output (bottom center) mostly follows the retrieved example but replaces the conditional with a generic one.

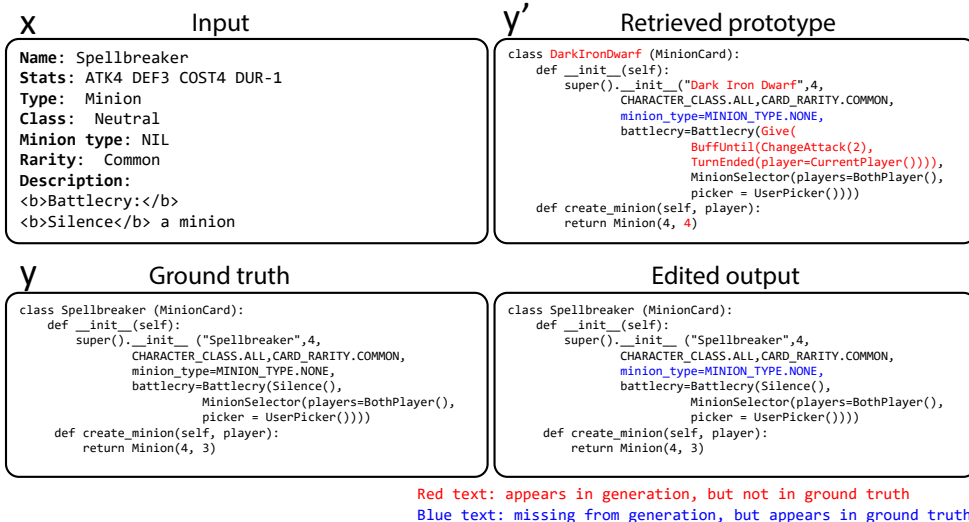

Figure 3. Example from the Hearthstone validation set (left panels) and the retrieved example used during prediction (top right). The output (bottom right) differs with the gold standard only in omitting an optional variable definition (minion_type).

| | BLEU | Accuracy |
|---|---|---|
| | AST based | |
| Abstract Syntax Network (ASN) [26] | **79.2** | **22.7** |
| Yin *et al*[41] | 75.8 | 16.2 |
| | Non AST models | |
| Retrieve+Edit (this work) | **70.0** | **9.1** |
| Latent Predictor Network [22] | 65.6 | 4.5 |
| Retriever [22] | 62.5 | 0.0 |
| Sequence-to-sequence [22] | 60.4 | 1.5 |
| Statistical MT [22] | 43.2 | 0.0 |

Table 3. Retrieve-and-edit substantially improves upon standard sequence-to-sequence approaches for Hearthstone, and closes the gap to AST-based models.

## 4.2 Hearthstone cards benchmark

The Hearthstone cards benchmark consists of 533 cards in a computer card game, where each card is associated with a code snippet. The task is to output a Python class given a card description. Figure 3 shows a typical example along with the retrieved example and edited output. The small size of this dataset makes it challenging for sequence-to-sequence models to avoid overfitting to the training set. Indeed, it has been observed that naive sequence-to-sequence approaches perform quite poorly [22].

For quantitative evaluation, we compute BLEU and exact match probabilities using the tokenization and evaluation scheme of [41]. Retrieve+Edit provides a 7 point improvement in BLEU over the sequence-to-sequence and retrieval baselines (Table 4.2) and 4 points over the best non-AST based method, despite the fact that our editor is a vanilla sequence-to-sequence model.

Methods based on ASTs still achieve the highest BLEU and exact match scores, but we are able to significantly narrow the gap between specialized code generation techniques and vanilla sequence-to-sequence models if the latter is boosted with the retrieve-and-edit framework. Note that retrieve-and-edit could also be applied to AST-based models, which would be an interesting direction for future work.

Analysis of example outputs shows that for the most part, the retriever finds relevant cards. As an example, Figure 3 shows a retrieved card (DarkIronDwarf) that functions similarly to the desired output (Spellbreaker). Both cards share the same card type and attributes, both have a battlecry, which is a piece of code that executes whenever the card is played, and this battlecry consists of modifying the attributes of another card. Our predicted output corrects nearly all mistakes in the retrieved output, identifying that the modification should be changed from ChangeAttack to Silence. The output

differs from the gold standard on only one line: omitting the line minion_type=MINION_TYPE.none. Incidentally, it turns out that this is not an actual semantic error since MINION_TYPE.none is the default setting for this field, and the retrieved DarkIronDwarf card also omits this field.

# 5 Related work

**Retrieval models for text generation.** The use of retrieval in text generation dates back to early example-based machine translation systems that retrieved and adapted phrases from a translation database [33]. Recent work on dialogue generation [40, 30, 37] proposed a joint system in which an RNN is trained to transform a retrieved candidate. Closely related work in machine translation [13] augments a neural machine translation model with sentence pairs from the training set retrieved by an off-the-shelf search engine. Retrieval-augmented models have also been used in image captioning [18, 24]. These models generate captions of an image via a sentence compression scheme from an initial caption retrieved based on image context. Our work differs from all the above conceptually in designing retrieval systems explicitly for the task of editing, rather than using fixed retrievers (e.g., based on lexical overlap). Our work also demonstrates that retrieve-and-edit can boost the performance of vanilla sequence-to-sequence models without the use of domain-specific retrievers.

A related edit-based model [14] has also proposed editing examples as a way to augment text generation. However, the task there was *unconditional* generation, and examples were chosen by random sampling. In contrast, our work focuses on conditional sequence generation with a deterministic retriever, which cannot be solved using the same random sampling and editing approach.

**Embedding models.** Embedding sentences using noisy autoencoders has been proposed earlier as a sentence VAE [5], which demonstrated that a Gaussian VAE captures semantic structure in a latent vector space. Related work on using the von-Mises Fisher distribution for VAE shows that sentences can also be represented using latent vectors on the unit sphere [9, 14, 38]. Our encoder-decoder is based on the same type of VAE, showing that the latent space of a noisy encoder-decoder is appropriate for retrieval.

Semantic hashing by autoencoders [16] is a related idea where an autoencoder's latent representation is used to construct a hash function to identify similar images or texts [29, 6]. A related idea is cross-modal embeddings, which jointly embed and align items in different domains (such as images and captions) using autoencoders [39, 2, 31, 10]. Both of these approaches seek to learn general similarity metrics between examples for the purpose of identifying documents or images that are semantically similar. Our work differs from these approaches in that we consider *task-specific* embeddings that consider items to be similar only if they are useful for the downstream edit task and derive bounds that connect similarity in a latent metric to editability.

**Learned retrieval.** Some question answering systems learn to retrieve based on supervision of the correct item to retrieve [35, 27, 19], but these approaches do not apply to our setting since we do not know which items are easy to edit into our target sequence $y$ and must instead estimate this from the embedding. There have also been recent proposals for scalable large-scale learned memory models [34] that can learn a retrieval mechanism based on a known reward. While these approaches make training $p_{\text{ret}}$ tractable for a known $p_{\text{edit}}$, they do not resolve the problem that $p_{\text{edit}}$ is not fixed or known.

**Code generation.** Code generation is well studied [21, 17, 4, 23, 1], but these approaches have not explored edit-based generation. Recent code generation models have also constrained the output structure based on ASTs [26, 41] or used specialized copy mechanisms for code [22]. Our goal differs from these works in that we use retrieve-and-edit as a general-purpose method to boost model performance. We considerd simple sequence-to-sequence models as an example, but the framework is agnostic to the editor and could also be used with specialized code generation models. Recent work appearing after submission of this work supports this hypothesis by showing that augmenting AST-based models with AST subtrees retrieved via edit distance can boost the performance of AST-based models [15].

**Nonparametric models and mixture models.** Our model is related to nonparametric regression techniques [32], where in our case, proximity learned by the encoder corresponds to a neighborhood,

and the editor is a learned kernel. Adaptive kernels for nonparametric regression are well-studied [11] but have mainly focused on learning local smoothness parameters rather than the functional form of the kernel. More generally, the idea of conditioning on retrieved examples is an instance of a mixture model, and these types of ensembling approaches have been shown to boost the performance of simple base models on tasks such as language modeling [7]. One can view retrieve-and-edit as another type of mixture model.

## 6 Discussion

In this work, we considered the task of generating complex outputs such as source code using standard sequence-to-sequence models augmented by a learned retriever. We show that learning a retriever using a noisy encoder-decoder can naturally combine the desire to retrieve examples that maximize downstream editability with the computational efficiency of cosine LSH. Using this approach, we demonstrated that our model can narrow the gap between specialized code generation models and vanilla sequence-to-sequence models on the Hearthstone dataset, and show substantial improvements on a Python code autocomplete task over sequence-to-sequence baselines.

**Reproducibility.** Data and code used to generate the results of this paper are available on the CodaLab Worksheets platform at `https://worksheets.codalab.org/worksheets/0x1ad3f387005c492ea913cf0f20c9bb89/`.

**Acknowledgements.** This work was funded by the DARPA CwC program under ARO prime contract no. W911NF-15-1-0462.

## Footnotes

[1] This expression is the conditional entropy $H(y \mid x', y')$. An alternative interpretation of $\mathcal{L}^*$ is that maximization with respect to $p_{\text{ret}}$ is equivalent to maximizing the mutual information between $y$ and $(x', y')$.

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
