[Supplementary Material]

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

# A  KL divergence between two von Mises-Fisher distributions

Here, we show that the KL divergence between two vMF distributions with the same concentration parameter $\kappa$ is proportional to the squared Euclidean distance between their respective direction vectors.

**Proposition 2.** *Let $\mu_1, \mu_2 \in \mathbb{S}^{d-1}$, then*

$$KL(vMF(\mu_1, \kappa)\|vMF(\mu_2, \kappa)) = C_\kappa \left\|\mu_1 - \mu_2\right\|_2^2.$$

*where $C_\kappa = \kappa \frac{I_{d/2}(\kappa)}{2I_{d/2-1}(\kappa)}$.*

**Proof**  Let $v \sim \text{vMF}(\mu, \kappa)$, then $\mathbb{E}(v) = \frac{I_{d/2}(\kappa)}{I_{d/2-1}(\kappa)}\mu$. Further, define $\text{vMF}(v; \mu, \kappa) = Z_\kappa^{-1}\exp(\kappa v^\top \mu)$ then,

$$
\begin{aligned}
KL(\text{vMF}(\mu_1, \kappa)\|\text{vMF}(\mu_2, \kappa)) &= \mathbb{E}[\kappa v^\top \mu_1 - \kappa v^\top \mu_2] \\
&= \kappa \mathbb{E}[v]^\top \mu_1 - \kappa \mathbb{E}[v]^\top \mu_2 \\
&= \kappa \frac{I_{d/2}(\kappa)}{I_{d/2-1}(\kappa)} \left(1 - \langle \mu_1, \mu_2 \rangle \right).
\end{aligned}
$$

If we let $\kappa \frac{I_{d/2}(\kappa)}{2I_{d/2-1}(\kappa)} = C_\kappa$, then $KL(\text{vMF}(\mu_1, \kappa), \text{vMF}(\mu_2, \kappa)) = C_\kappa \|\mu_1 - \mu_2\|_2^2$.  $\square$

# B  Model details

## B.1  Editor

Recall that the editor takes an input $x$ and a retrieved example $(x', y')$, and outputs a target sequence $y$. Our model is a standard seq2seq model, where inputs $(x, x', y')$ are encoded using separate LSTMs, and a decoder uses both the encoded vectors and a standard attention mechanism over the inputs $(x, x', y')$ to generate the target sequence $(y)$. For completeness, we describe the model and any non-standard components below.

The encoder is a 2-layer bi-directional LSTM [8]. Each component of the input structure such as the context $x$ and retrieved example $(x', y')$ is LSTM-encoded separately, and an additional linear layer is used to combine the final hidden states.

We employ this architectural scheme for the VAE encoder $\mu(x, y)$, context encoder $\mu(x)$, and edit model encoder.

The decoder of the editor is a 4-layer LSTM which attends to its input as in [3], where the top layer LSTM's hidden states are used to compute attention over the encoder's LSTM states. This decoder architecture is used in the editor and used in the VAE decoder without the attention mechanism.

The edit model decoder additionally makes use of a copying mechanism, which enables the neural editor to copy words from its input (in our case, the retrieved example) instead of generating them, by emitting a specialized copy token. This is critical for the slot-filling behavior of our model when making the retrieved function relevant for the current context. The copy mechanism follows the overall approach of [12], with a simplified copy embedding which is defined by position, rather than the hidden states of the encoder. The full details of this copy mechanism are below

For the edit model, also we augment the dataset by replacing the training example $(x, x', y') \rightarrow y$ with the identity map $(x', x', y') \rightarrow y'$ with probability $0.1$. Removing this augmentation did not substantially change the results, but did increase sensitivity to the choice of the number of training epochs.

## B.2  Copying

The base vocabulary is extended with a number of *copy tokens* (300 in our implementation, though this number need not be larger than the largest number of tokens in any input sequence.) Each copy

token corresponds to the action of copying the word at a particular position in the input sequence. The embedding of a word then becomes the concatenation of the vector corresponding to its token in the base vocabulary with the vector of the copy token which corresponds to the word's position in the input sequence.

At both train and test time, the decoder computes a soft-max over all tokens in the vocabulary (including copy tokens). At train time, the probability of generating a word in the target sequence, and analogously its log-likelihood, is defined as the sum of the probability of generating the base word, and the probability of generating the copy token corresponding to the instance of the word in the input sequence, if such an instance exists. The target soft-max at each time-step places a probability mass of $1$ on both the target word and and the copy token corresponding to its instance in the input sequence, if it exists. At test time, probability mass placed on copy tokens is instead transferred over to their corresponding words, if they exist in the vocabulary as base words. When a copy token is emitted and then conditioned upon, it is replaced with the token of the word to which it corresponds in the input sequence, if it is in the vocabulary.

### B.3 von Mises-Fisher distribution

The von Mises-Fisher distribution with mean $\mu \in \mathbb{R}^d$ and concentration $\kappa \in \mathbb{R}_{\geq 0}$ has a probability density function defined by

$$\text{vMF}(\boldsymbol{x}; \mu, \kappa) = \mathcal{C}_d(\kappa) \exp(\kappa \mu^T \boldsymbol{x})$$

with normalization constant

$$\mathcal{C}_d(\kappa) = \frac{\kappa^{d/2-1}}{(2\pi)^{d/2}\mathcal{I}_{d/2-1}(\kappa)}$$

where $I_\nu$ denotes the modified Bessel function of the first kind of order $\nu$.

## C  Github dataset construction

We scraped the Google Bigquery dataset using the following regex query designed to retrieve all python files of at least 100 bytes containing at least one function with a restructured text (reST) style docstring.

```
SELECT *
FROM [bigquery-public-data:github_repos.files] f JOIN [bigquery-public-data:github_repos.contents
ON f.id = c.id
WHERE
  REGEXP_MATCH(c.content,r'"""[\s\S]*?returns:[\s\S]*?"""')
HAVING RIGHT(f.path,2) = 'py' AND c.size > 100
LIMIT 5000000
```