[Reviews · NeurIPS 2018]

Reviewer 1



This paper addresses the task of highly structured output prediction in the contexts of, here, source code generation. The authors propose a two-step process for this, consisting of a retrieval model that retrieves the closest seen training instance to the test instance, and an edit model, that edits the retrieved training instance. Quality: - The authors approach the relatively recently proposed task of generating Python source code. Unlike others, they work in a setting with less supervision, where they assume no access to the Abstract Syntax Tree. - The key innovation here is the task-specific retrieval model, trained using a variational autoencoder that is then further fine-tuned such that the output distribution of the learned encoder resembles the oracle encoder. I like this idea and the proposed model a lot and can see how it would be useful beyond the immediate scope of the paper. - It is clear how the proposed method differs from previous work, and related work is adequately cited - What wasn't clear to was why the retriever only retrieves one training example to be edited per testing instance. This seems to rely very heavily on the retriever's performance and also on the training set containing one perfectly related training instance. Wouldn't it be better to retrieve several and then try and then have the edit model merge them as well? - The authors could have shown more ablation studies for their results. They make several assumptions in their approach for which detailed results are not shown, e.g. the encoding prior (Section 3.1.1) - They assume that there is an oracle editor, but don't describe how the oracle decoder is constructed from the training data. Also, they didn't discuss the dependence of their approach on the oracle editor or how could train the model in a scenario without access to an oracle editor. Clarity: - The submission is overall well-written and well-motivated. The figures provided nicely illustrate the approach. - In several places in the paper, abbreviations are used without being introduced (e.g. "AST" in abstract) and variables are introduced after formulae they are used in, particularly in Section 3. Small rewrites could make the paper more readable in those places. Originality: - This work is a novel combination of existing techniques, combining variational autoencoders, joint training, and sequence models. Previous work using retrieval as part of their approach saw this mainly as a black-box component and did not try to jointly train the retrieval component with the main model. Significance: - As already stated above, I believe the general idea as well as the model proposed here will be used beyond the immediate scope of this work (Python code generation). ================= Thanks to the authors for clarifying my questions. My remaining concerns are addressed and I would be happy to see this paper appear at NIPS. In light of this, I increased my score by 1.

Reviewer 2



Thank you very much for the response. This is an interesting paper to see in the conference. I'm still wondering if the model is relying on the repetitive patterns in the small training examples and curious to see your clarifications. ================================ The authors propose a conditional text generation model which retrieve similar example from training data based on input and generate output conditioned on the input and the retrieved example. They kept their decoder to be a standard sequence-to-sequence model with attention and coping mechanism and put more effort into a retriever network. Their retrieval network is trained in two steps. They first trained a hyperspherical VAE model where its recognition model project an input and output pair to a point in the hypersphere. Then they trained a retrieval network which only has access by encouraging the model to point to the same place as the recognition model trained in the first step. They applied their model on code generation task and code completion task and obtained substantial improvements over baseline models. The paper is clearly written. They explained their motivations and ideas well and demonstrated solid results with interesting examples from their model. Although there are many models which try to generate text with retrieve and edit framework, their model is different in a sense that they learned a task-dependent model for retrieval instead of using existing search engines or other heuristics. To make the paper stronger, it will be great to see code generation experiments with different baselines, e.g. InputReader etc. For me, it is not clear why the InputReader is not enough as it has to contain the enough information to query. Also, it is worth discussing if the same model is applicable for tasks with large corpora such as translation and dialogue generation. Although the results are solid in this paper, their dataset is quite small (only 500 training examples) and there will be a lot of common patterns in examples. The model is suitable for this kind of dataset as they need to run a search over all examples and it may suffer from searching over a large corpus.

Reviewer 3



This paper presents a creative new approach to structured prediction. Instead of using a parametric model to directly form a distribution on output given input, the proposed model is comprised of a retriever and editor: the retriever finds a training point that is most easily edited to form the correct output, and the editor performs the editing. This work is based on [13] where a non-conditional generative retrieve and edit model was proposed, but adds substantial and important new ideas in adapting this approach to conditional generation. Also interesting is the novel training technique and the component approximations, which add intuition and motivation. Instead of attempting to jointly learn a parameterized retriever and editor, this paper takes what appears at first to be a simple pipelined approach. However, the authors demonstrate that the VAE-style objective for training the retriever is a formal lower bound on an objective that corresponds to training the retriever in the context of an optimal editor. In the end, the pipelined nature of the overall approach means it is extremely simple to plug in additional editors -- this whole technique is easily ported to many tasks and models. Overall, these ideas are new and interesting and will go on to inspire lots of future work. Further, because of its simplicity, I expect this approach will be very useful in practice. Even without positive results I would advocate for this paper -- and the results are strong on two difficult tasks! Misc: -line 65: What about stochastic EM? Wouldn't this be tractable alternative for joint learning? -Even if we could tractably sum over training data, might we still see better performance from using oracle training for the retriever? -line 142: But the true data generating distribution doesn't depend on (x', y'). Have I misunderstood something? Or are we assuming our dataset was produced by an editing process? This seems like a critical point in understanding how the VAE objective is a lower bound on the optimal scenario for retrieval. Probably worth expanding a bit here.